# A Narrative Review: The Role of NETs in Acute Respiratory Distress Syndrome/Acute Lung Injury

**DOI:** 10.3390/ijms25031464

**Published:** 2024-01-25

**Authors:** Xinyu Zhou, Jiajia Jin, Tangfeng Lv, Yong Song

**Affiliations:** Department of Respiratory and Critical Care Medicine, Jinling Hospital, Affiliated Hospital of Medical School, Nanjing University, Nanjing 210093, China; 522023350250@smail.nju.edu.cn (X.Z.); jjj_0817@163.com (J.J.)

**Keywords:** ARDS, ALI, NETs, NETosis, COVID-19, inflammatory storm, immune balance

## Abstract

Nowadays, acute respiratory distress syndrome (ARDS) still has a high mortality rate, and the alleviation and treatment of ARDS remains a major research focus. There are various causes of ARDS, among which pneumonia and non-pulmonary sepsis are the most common. Trauma and blood transfusion can also cause ARDS. In ARDS, the aggregation and infiltration of neutrophils in the lungs have a great influence on the development of the disease. Neutrophils regulate inflammatory responses through various pathways, and the release of neutrophils through neutrophil extracellular traps (NETs) is considered to be one of the most important mechanisms. NETs are mainly composed of DNA, histones, and granuloproteins, all of which can mediate downstream signaling pathways that can activate inflammatory responses, generate immune clots, and cause damage to surrounding tissues. At the same time, the components of NETs can also promote the formation and release of NETs, thus forming a vicious cycle that continuously aggravates the progression of the disease. NETs are also associated with cytokine storms and immune balance. Since DNA is the main component of NETs, DNase I is considered a viable drug for removing NETs. Other therapeutic methods to inhibit the formation of NETs are also worthy of further exploration. This review discusses the formation and mechanism of NETs in ARDS. Understanding the association between NETs and ARDS may help to develop new perspectives on the treatment of ARDS.

## 1. Introduction

Acute respiratory distress syndrome (ARDS) is the acute onset of respiratory disease characterized by bilateral pulmonary edema and hypoxemia of noncardiac origin caused by damage to the pulmonary endothelial barrier and excessive permeability of alveolar capillaries, which is clinically manifested as bilateral pulmonary infiltrates and respiratory failure [1]. The Berlin definition classifies ARDS as mild, moderate, or severe based on the level of pulmonary oxygenation, and removes the definition of acute lung injury (ALI) [2]. The most common causes of ARDS include pneumonia and non-pulmonary sepsis, as well as the aspiration of the stomach contents, trauma, pancreatitis, burns, inhalation injury, drug overdose, multiple blood transfusions or shock, e-cigarettes, chemotherapy and immunotherapy (including checkpoint inhibitors), etc. [3,4]. A 2016 observational study covering 459 intensive care units (ICU) in 50 countries showed that 10% of ICU patients and 23% of mechanically ventilated patients had ARDS, with a 28-day mortality rate of 35% and a mortality rate of more than 40% for patients with severe ARDS [5]. In addition to a high incidence and mortality, patients with ARDS may have a sequelae of physical, cognitive, and mental health conditions after recovery [3]. Alveoli neutrophil infiltration, which causes persistent inflammation, is one of the symptoms of ARDS [6] (Figure 1).

Neutrophils, a major component of innate immunity and the host’s first line of defense against infectious pathogens, are recruited at the onset of infection and destroy pathogens through mechanisms such as phagocytosis, degranulation, production of reactive oxygen species, release of antimicrobial peptides, and the recently discovered formation of neutrophil extracellular traps (NET) [7]. NETs are produced by activated neutrophils and consist of DNA, histones, granular proteins such as neutrophil elastase, cathepsin G, and myeloperoxidase [8]. The process of NET formation by neutrophils, known as NETosis, is a novel mechanism of programmed cell death of neutrophils, and can be divided into NADPH oxidase 2 (NOX)-dependent and independent NETosis [9] (Figure 2A). NETs can be induced by IL-8, LPS, etc., and have strong antibacterial effects [8]. In addition, NETs can also mediate tissue damage [10], cancer [11], inflammation [12], and autoimmune diseases [13]. Recently, the association between NETs and ARDS has been confirmed by a number of studies. This review describes the interaction between NETs and ARDS, and discusses the relationship between NETs and inflammatory storms and immune balance, while focusing on the production of NETs and the effect on the progression of ARDS disease.

## 2. Evidence on the Influence of NETs on ARDS

There is increasing evidence that NETs are involved in ARDS, and NETs have been shown to play an important role in both infectious and aseptic ALI, including direct and indirect ALI [14,15]. Compared with healthy controls, more NETs were detected in the bronchoalveolar lavage fluid (BALF), tracheal aspiration fluid, and plasma of ARDS patients [16,17,18]. This may be due to significantly reduced neutrophil apoptosis, an extended neutrophil lifespan, and the greater ability of ARDS patients to form NETs [16,18]. At the same time, Murielle Gregoire et al. showed in their observational study that bronchoalveolar lavage fluid in ARDS patients can induce pulmonary epithelial cell damage by producing more NETs in the neutrophils from both healthy donors and ARDS patients [16]. In addition to adult ARDS, a study involving 77 children with airway aspiration also suggested that the level of NETs in the airway aspiration of children with pediatric ARDS (PARDS) was significantly higher than that in children without PARDS, and the amount of NETs formed was negatively correlated with the number of ventilator-free days [19].

Sepsis and bacterial pneumonia are common causes of ARDS, and NETs were increased in patients with sepsis or infectious pneumonia-induced lung injuries [15,17,20,21,22,23,24]. Reducing the formation of NETs by decreasing PAD4 or increasing the degradation of NETs by increasing DNase I can reduce the level of NETs, and thus relieve lung inflammation and lung injury [20,21,23,24]. Interestingly, a laboratory study by Emma Lefrancais and colleagues discovered that mice with NETosis damage showed impaired bacterial clearance in addition to reduced lung damage, and that the beneficial effects gained through decreasing NET formation were counteracted by the increased bacterial load and inflammation [20]. These results suggest that NETs have both antibacterial and proinflammatory effects in sepsis and pneumonia-associated ARDS. The increase in NET level was also found in patients with lung injuries caused by other infectious diseases, such as malaria [25] and H1N1 [26], suggesting that the formation of NETs is related to the pathogenesis of ALI/ARDS.

Transfusion-related acute lung injury (TRALI) is a type of ALI that occurs within 6 h of a transfusion and cannot be explained by other risk factors for ALI [27]. Studies have shown that NET levels are elevated after a TRALI [28,29]. The release of NETs is induced by activated platelets in the lung, dependent on the canonical Raf/MEK/ERK signaling pathway [30]. However, another study by Thomas and colleagues found that in addition to platelet-induced NETosis, anti-neutrophilic antibodies promote NET formation through a FcγR binding-dependent mechanism [28]. In terms of the pathogenesis of ventilator-induced lung injury (VILI), existing evidence suggests that NETs are released during VILI [31,32,33]. Jan Rossaint’s team found that NET formation in VILI is platelet-dependent and requires the simultaneous stimulation of the integrin and G-protein-coupled receptors on neutrophils [31]. Li and colleagues found that the release of NETs in VILI was related to the TLR4 signaling pathway [33].

Furthermore, an increase in NETs in the lungs was also found in indirect lung injuries caused by the severe injury of other tissues and organs [17,34,35,36]. The occurrence and development of acute lung injuries caused by hip necrotizing fasciitis, skin burns, acute kidney injury, and intestinal ischemia/reperfusion (I/R) are all related to the formation and release of NETs [17,34,35,36], which can disrupt the pulmonary microvascular endothelial cell barrier function and aggravate lung injury [35]. The degradation and inhibitory treatment of NETs can reduce NET formation, tissue inflammation, and pathological injury in the lungs, and have a protective effect on ALIs [34,35,36].

Moreover, many studies have shown that NETs in the plasma [18,20,24,28] or tracheal aspirated fluid [19,23] of ARDS patients are positively correlated with the severity and mortality of ARDS, and can be used as an indicator to predict the prognosis of ARDS patients [15]. The amount of NETs formed can change dynamically across the clinical progression of ARDS [17]. Masahiro Ojima et al. observed in the tracheal aspirates of three patients with different conditions that the amount of NETs decreased in patients with improved conditions, the formation of NETs increased in patients with aggravated conditions, and the persistent expression of NETs in patients who failed to recover from ARDS and eventually died of respiratory failure [17]. As existing studies only correlate elevated levels of NETs with the progression of ARDS, without indicating specific signaling pathways and causality, NETs as a potential therapeutic target or prognostic marker of ARDS need further investigation.

## 3. The Mechanism of NETs Influencing ARDS

NETs may cause damage to lung epithelial and endothelial cells through their components, including histones, myeloperoxidase (MPO), NE, DNA, and cathepsin G [15,37,38] (Figure 2B,C).

### 3.1. Histones

Usually located in the nucleus, histones are components of chromatin that bind to and regulate the expression of DNA; at the same time, histones are important components of NETs, accounting for 70% of all NET-related proteins, have host defense functions, and can promote inflammatory responses [39,40]. The catalytic conversion of histone arginine to citrulline mediated by peptidyl arginine deiminase 4 (PAD4) can change the interaction between histone molecules, reduce the stability of histone, and promote the decondensation of heterochromatin, which is a necessary condition for the formation of NETs [41,42,43,44,45]. Reduced NETs and reduced lung injury due to PAD4 inhibitor treatment or PAD4 gene defects also demonstrate the critical role of PAD4 [36,43,46,47]. However, other studies have shown that NETs release in the presence of PAD4 inactivation [48,49,50]. In addition to PAD4, neutrophil elastase (NE) can also cleave histones during NET formation through nuclear translocation [51]. Kolaczkowska et al. found that after DNA enzyme treatment, the cytotoxicity of NETs did not disappear, and there were still a large number of active histones causing tissue damage [47]. Saffarzadeh et al. also confirmed this, and found that the use of protein antibodies reduced the cytotoxicity of NETs, revealing the histone-mediated cytotoxicity of NETs [38].

Histones can cause endothelial cell damage, alveolar bleeding, and thrombosis [29,52,53]. Axelle Caudrillier et al. found that NETs were present in large numbers in TRALI patients and mouse models, and that NETs damaged endothelial cells, increasing their permeability, while histone-blocking antibodies were protective against pulmonary endothelial injury in TRALI [29]. The primary mechanism of histone cytotoxicity to endothelial cells is calcium influx due to plasma membrane disruption, which has been reported by Simon T. Abrams et al. [53]. The pathological examination in the experiments by Abrams et al. showed that extracellular histones can cause intra-alveolar hemorrhages and microvascular thrombosis, which is also reflected in a study by Jun Xu and colleagues [52,53]. Abrams et al. also mentioned that the formation of pulmonary thrombosis strongly supports the involvement of histone-triggered coagulation activation in lung injury, which is in line with the experimental results reported by the Tobias A. Fuchs group, who found that histone activates αIIb3 integrin on platelets and the recruitment of fibrinogen in order to induce platelet aggregation [53,54]. These pieces of evidence emphasize that the main mechanisms of histone-induced lung injuries are endothelial cell injury and coagulation activation. Previous studies have shown that the TLR family is involved in histone damage mechanisms, such as TLR2 and TLR4, which mediate the process of histone activating platelets to release thrombin [55], and extracellular histones activating NLRP3 inflammasome via TLR9 signaling [56]. However, TLR2 and TLR4 neutralizing antibody treatments did not mitigate histone-induced endothelial cell damage [53], suggesting that TLRS is not the primary pathway mediating histone cytotoxicity for endothelial cells. Interestingly, histones have been shown to induce the formation of NETs in the same way as PMA [53], so it is likely that histones and NETs will form a vicious cycle and further aggravate lung injury.

### 3.2. NE

Neutrophil elastin (NE) is the highest non-histone protein in NETs [39]. As a class of serine proteases, NE is capable of directly degrading virulence factors and participating in the proteolytic process of chemokines, cytokines, and receptors, and is the major source of proteolytic activity of NETs [57]. Just like histones, NE is also associated with the endothelial cytotoxicity of NETs [37,58]. NE mediates neutrophil-induced tissue damage and effectively degrades the extracellular matrix [59]. NE also degrades the endothelial cytoskeleton by affecting the function of E-cadherin and VE-cadherin, thereby undermining the integrity of the alveolar–capillary barrier [60,61]. Sivelestat is a selective inhibitor of NE. Suzuki et al. found that Sivelestat mitigated LPS-induced lung endothelial cell injury, reduced thrombomodulin, and syndecan-1 degradation, and also protected endothelial glycocalyx [58]. Okeke et al.‘s finding that Sivelestat alleviates NET-induced injuries of human umbilical vein endothelial cells also reaffirms the cytotoxicity of NE on endothelial cells [62]. In addition, Okeke et al. inhibited NET formation using Sivelestat, indicating that NE plays an important role in NET formation [62], which is in-line with the previously mentioned results that NE affects NET formation through the nuclear translocation cleavage of histones [51]. However, it has also been noted that NE inhibition alone is not sufficient to inhibit NET release [63] or inhibit NET cytotoxicity [38]. Therefore, the specific role of NE in NETosis needs further investigation, including into its possible cytotoxicity independent of enzyme activity [38].

### 3.3. DNA

DNA is the major structural component of NETs, and transient treatment with deoxyribonuclease (DNase) can cause the disintegration of NETs, indicating that DNA maintains the integrity of NETs [8]. In addition to this, the DNA in NETs has been shown to be associated with coagulation. NETs provide a scaffold for platelet binding and stimulate platelet aggregation. Interactions between NETs and platelets are regulated in a number of ways, including the direct interaction of platelets with the DNA in NETs [64]. Travis J. Gould et al.’s experimental results showed that NETs in patients with sepsis increased thrombin production through intrinsic pathways; that is, cell-free DNA in NETs mediated thrombin production through coagulation factor XII- or coagulation factor XI-dependent coagulation pathways [65]. In addition, several experiments have shown that the procoagulant effects of NETs was weakened after DNase treatment, which further confirmed the link between the DNA in NETs and coagulation [65,66]. Christian Lood et al. mentioned that, in addition to chromosome DNA, NETs can enrich mitochondrial DNA (mtDNA), and the released oxidized mtDNA has a highly pro-inflammatory effect, which can stimulate type I interferon (IFN) signaling through the DNA sensor STING pathway [67]. At the same time, another study showed that the NETs formed after major trauma and surgery was only mtDNA, which was a marker of the enhanced activation of innate immunity [68], which was consistent with the conclusion proposed by S Yousef et al. that NETs containing mtDNA are an important part of innate immunity [69]. Additional mtDNA proinflammatory pathways were found through the stimulation of the CpG DNA sensor TLR9 [70] and the activation of the NLRP3 inflammatory pathway on immune cells [71]. However, in addition to participating in the proinflammatory effects of mtDNA, TLR9 is also a pathway through which mtDNA stimulates neutrophils to release NETs [8,10,72,73]. Therefore, releasing mtDNA and NETs may form a vicious cycle, constantly aggravating tissue damage and inflammatory reactions.

### 3.4. MPO

Myeloperoxidase (MPO), another component of NETs, is also a cause of NET-mediated cytotoxicity [8,37,38,39]. MPO converts hydrogen peroxide into hypochlorous acid and produces ROS, which can help kill bacteria or cause tissue damage [38,74]. One study showed that MPO released after secondary necrosis of neutrophils induced DNA strand breaks in lung epithelial cells, resulting in lung epithelial cell damage [8]. Saffarzadeh et al. demonstrated that MPO was associated with NET cytotoxicity by reducing the cytotoxicity of NETs to epithelial cells through preincubation with MPO inhibitors [38]. A study by Kathleen D. Metzler and her team showed that MPO is necessary for the formation of NETs [75]. Tokuhiro et al. demonstrated that MPO was involved in the formation of NETosis and NETs through oxidized phospholipids, oxidized phospholipids, and NE synergistically promoted chromatin decompaction [49]. This supports the importance of MPO and suggests the potential of MPO as a therapeutic target for blocking NET formation in lung injuries.

In conclusion, NETs damage lung tissue in various forms through their components and promote the occurrence and development of ARDS. Histone, NE, and MPO can damage lung endothelial cells and epithelial cells, and destroy the barrier. Histones and DNA also promote coagulation and mediate the formation of immune thrombus interacting with inflammatory responses, becoming key events in the development of ARDS [60]. In addition, DNA can activate STING-IFN, TLR9, NLRP3, and other pathways, playing a pro-inflammatory role. Other studies have shown that LPs-mediated NETs can induce the pro-death of macrophages and thus regulate the inflammation of ARDS [33]. NETs mediated by free fatty acids can induce ALI by mediating dendritic cell activation and T cell differentiation [76]. NETs play a key role in sepsis-associated lung injury by inducing the m6A modification of GPX4 through the up-regulation of methyltransferase-like 3 and subsequently inducing ferroptosis in alveolar epithelial cells [24]. A more interesting phenomenon is that important components of NETs, including histones, DNA, NE, and MPO, all play important roles in the formation of NETs. In other words, in the process of lung injury, NETs and their components may be able to form a vicious circle, resulting in an uncontrolled inflammatory response and increasing tissue damage. Therefore, the mechanism of NET components regulating ARDS needs to be further studied in detail, and the possibility of blocking the cytotoxicity of NETs and the release of NETs by blocking these components of NETs is also worth exploring.

## 4. The Mechanism of NETs Production in ARDS and the Substances That Affect the Production

The production and effect of NETs in ARDS have been confirmed by many studies. The mechanism of NET production in ARDS and the substances that affect the formation and release of NETs are also under constant exploration (Table 1).

NETosis can be divided into NADPH oxidase 2 (NOX)-dependent and independent NETosis [9] (Figure 2A). PKC can activate NOX through the Raf-MEK-ERK signal transduction pathway to produce reactive oxygen species (ROS) [28,77]. ROS triggers the separation, release, and migration of neutrophil elastase (NE) and myeloperoxidase (MPO) to the neutrophil nucleus. NE triggers histone degradation, and MPO collaborates with NE to promote chromatin decondensation. Activated peptidylarginine deiminase 4 (PAD 4)-mediated histone citrullination leads to chromatin decondensation [41,77]. The process of the formation and release of NETs includes nuclear and granule membrane dissolution, chromatin decondensation, and plasma membrane breakdown [9]. However, there is also NETosis in which neutrophils remain intact, and calcium-mediated and platelet-stimulated Toll-like receptor 4 (TLR4)-mediated NETosis have also been demonstrated to be ROS-independent [77].

PAD4 is essential for mediating NET formation [41], and PAD4 activated by PKC or the influx of Ca^2+^ is capable of citrullinating histones [9,78], so it is possible to regulate the production of NETs by regulating PAD4. A study in a mouse model of LPS-induced endotoxin reduced NET formation by treatment with the PAD2/PAD4 inhibitor YW356, thereby attenuating acute lung injury in mice [79]. PAD4 inhibitors also prevent NETosis in ALI caused by skin chemical burns or intestinal ischemia/reperfusion [35,36]. Interestingly, not all NET production is blocked by PAD4 inhibitors; for example, PAD4 inhibition does not affect cholesterol crystal-induced NET formation [13]. A variety of factors, including ionophores, bacterial products, and an increase in extracellular pH, can trigger the PAD4 enzymatic function in neutrophils and then promote the formation of NETs [80]. Moreover, both cold-inducible RNA-binding protein (CIRP) and miR-155 can also increase the production of NETs by increasing the expression of PAD4 [21,81]. In addition to mediating histone citrullination, PAD4 is also involved in NF-κB activation and IL-1β release. IL-1β triggers the cleavage of gasdermin D by serine proteases, leading to pore formation and NET formation [82]. Interestingly, in addition to PAD4-mediated histone citrullination, histone deacetylases (HADCs) also play a key role in the formation of NETs by reducing the acetylation of histone H3 to ensure the activation of PAD4 and NETosis [83].

Existing studies have shown that ROS plays an important role in the formation of NETs. LPS activates platelets and then promotes NET release through ROS-dependent classical NETosis or ROS-independent early/rapid NETosis, in which the classical pathway plays a dominant role and is regulated by IRF-1 [84]. Free fatty acids (FFA) have also been shown to activate the NOX pathway to promote ROS production and thus induce NETs production by neutrophils, while FFA can also induce NET formation via the p38 and JNK pathways [36]. The Raf-MEK-ERK pathway, which activates NADPH oxidase, is also involved in the formation of NETs through the up-regulation of antiapoptotic proteins [30]. In addition, there are other signaling pathways involved in NETosis, such as the lipoxin pathway. The lipoxin receptor Fpr2 and its ligands modify the calcium flux in neutrophils, thereby participating in citrullination and ROS production, and they regulate neutrophilic apoptosis [20]. And in TRALI, miR-144 activates the NF-κB/CXCR1 signaling pathway by inhibiting the expression of KLF2 to promote the formation of NETs [85].

Platelets are also associated with the formation of NETs [60,86]. Activated platelets induce neutrophil-forming NETs [87]. A recent study demonstrated that platelet-specific receptor glycoprotein (GP)VI, which mediates deleterious platelet activation, is required for NETosis in an LPS-induced ALI model, and is located early in thromboinflammation-driven ARDS/ALI [88]. Targeting platelet activation reduces the production of NETs in the TRALI model [29], and Jan Rossaint et al. demonstrated the platelet-dependent formation of NETs during VILI [31]. The mechanism of platelet-induced NETs is through TLR4. TLR4 on platelets can recognize and bind the TLR4 ligands of neutrophils, enabling neutrophils to be activated and release NETs [89]. Haosi Li and his team demonstrated that TLR4-knockout mice formed significantly fewer NETs during VILI than wild-type mice [33], confirming that TLR4 influences the production of NETs during lung injury. The porogenic protein gasdermin D (GSDMD) also plays a crucial role in the formation of NETs, and lysed DSDMD is able to form pores in the granular and plasma membranes, allowing NE to enter the cytoplasm and NETs to be released outside the neutrophils [90].

Various cytokines are involved in NETosis in ALI/ARDS, such as IL-8, which has been found to be associated with NET concentration in bronchoalveolar lavage fluid (BALF) in ARDS patients [91], and IL17A, which may synergistically induce NET production through STAT3 activation with other cofactors [92]. Jan Rossaint et al. found that integrins and G-protein coupled receptors on neutrophils are required for NET formation during VILI, and that blocking CCL5/CXCL4 heterodimers blocks NET formation [31]. Complement and microRNAs (miRs) both regulate NETosis, and the inhibition of complement activation can reduce endothelial injury, platelet retention, and NET release in TRALI [93], while miR144 and miR155 have been shown to promote the formation of NETs [81,85]. And in ARDS, up-regulated PD-L1 inhibits neutrophil autophagy through the PI3K/Akt/mTOR pathway, thereby maintaining the release of NETs [94]. What is more, the interferon inducer poly I:C may induce ALI and the formation of NETs by activating P38 MAPK and reducing the expression of claudin-5 [95].

**Table 1 ijms-25-01464-t001:** Substances that affect the production of NETs.

Correlates		Effect	References
PAD4		Mediates histone citrullination, be involved in NF-κB activation and IL-1β release, PAD4 inhibitors reduce the formation of NETs, and PAD4 enzymes promote the production of NETs.	[35,36,79,80,82]
ROS		Participates in classic NETosis, and triggers the separation, release, and migration of NE and MPO to the neutrophil nucleus.	[9,77,84]
Platelets		Activate neutrophils via TLR4 to release NETs.	[29,31]
Signaling pathways	Raf-MEK-ERK pathway	Activates NADPH oxidase and upregulates anti-apoptotic proteins, promoting the formation of NETs.	[30]
	Lipoxin pathway	Regulates the calcium flux of neutrophils, participates in citrullination and ROS production, and regulates neutrophil apoptosis and promotes NET formation.	[20]
	NF-κB/CXCR1 pathway	Activation promotes the formation of NETs.	[85]
Cytokines and others	IL-8	Triggers the cleavage of gasdermin D by serine proteases, leading to pore formation and NET formation, the content was correlated with the concentration of NETs in BALF.	[82,91]
	Gasdermin D (GSDMD)	Forms pores in the granular and plasma membranes, allowing NE to enter the cytoplasm and NETs to be released outside the neutrophils.	[90]
	IL17A	Induces NET production through STAT3 activation with other cofactors.	[92]
	Integrins and G-protein coupled receptors	Necessary for NET formation during the VILI process.	[31]
	CCL5/CXCL4 heterodimers	Blocking inhibits the formation of NETs.	[31]
	Complement	Inhibition of activation reduces the release of NETs.	[93]
	miR144, miR155	Promotes NET formation.	[81,85]
	PD-L1	Inhibits autophagy of neutrophils and maintain the release of NETs through the PI3K/Akt/mTOR pathway.	[94]
	Interferon inducer poly I:C	Induces the formation of NETs via activation of P38 MAPK and decreased expression of the tight-linking protein claudin-5.	[95]

## 5. Relationship between the Inflammatory Storm and NETs

Cytokine storms are a syndrome involving the overproduction of inflammatory cytokines and overactivation of immune cells, covering various events that may eventually lead to multiple organ failure and death, such as acute respiratory distress syndrome (ARDS) [96,97]. The link between cytokine storms and ARDS has been established, and NETs also play a role. What is more, the formation of NETs can induce macrophages to secrete IL-1β, which can further induce the formation of NETs [98]. Several studies have suggested that cytokines are one of the triggers of NETosis [99,100], and cytokine storms can induce the production of NETs [101,102]. Proinflammatory cytokines, including IFN-γ, TNF-α, and GM-CSF, can regulate the expression of chemokine receptors (CRs), and CRs induced in vitro can regulate the release of NETs from lung-recruited neutrophils [103], which may be one of the pathways by which cytokines regulate the formation of NETs. At the same time, NETs can also trigger cytokine storm formation [77]. The formation of NETs can induce the production of pro-inflammatory cytokines, and NETs can amplify inflammation by promoting the production of cytokines/chemokines, thus causing cytokine storms [104,105]. Yun Young Lee et al. found in their clinical trial that NF-κB activity and cytokine secretion decreased after treatment with long-acting DNaseⅠ in SARS-CoV-2 [106]. Similarly, research by Emeka B. Okeke’s team in a mouse model of endotoxin shock found that inhibiting NE in NETs reduced systemic proinflammatory cytokine levels in mice [62]. These findings confirm the link between NETs and cytokines, and that anti-NET therapy can reduce excessive inflammation and excessive immune response in ARDS.

## 6. Correlation between Immune Balance and Regulation in ARDS and NETs

In ALI, NETs can promote the activation and transformation of lung macrophages into pro-inflammatory M1 types, while M1 type macrophages can promote the infiltration of neutrophils in the lungs and exacerbate the production of NETs. The two cooperate to cause the disorder of immune cells and aggravate tissue injury [107]. During influenza virus infections, the recruitment and activation of innate immune cells such as neutrophils and macrophages are out of control, and a large number of neutrophils accumulate and release NETs in the lungs, causing damage and leading to ARDS [103]. Similarly, the ratio of neutrophils to lymphocytes increased in patients with severe COVID-19, and the neutrophil subsets of patients with severe COVID-19 were more active in producing NETs, which would aggravate the inflammation of ARDS [108,109]. In line with this, multiple studies have shown that immunomodulatory therapy is effective against COVID-19 [105], and targeting innate immune-related factor Bruton’s tyrosine kinase can reduce centrocyte infiltration and NET secretion, and alleviate lung injury [110]. These findings once again confirm the contribution of inflammation and immune disorders in the formation and development of ARDS, in which NETs are an important factor.

## 7. Production and Role of NETs in COVID-19 ARDS

Coronavirus disease 2019 (COVID-19), caused by severe acute respiratory syndrome coronavirus 2 (SARS-CoV-2), is a global pandemic infectious disease that can affect multiple systems, mainly the respiratory system. Acute lung injury (ALI) and acute respiratory distress syndrome (ARDS) are its most severe pulmonary manifestations [111]. Elevated levels of NETs have been found in patients with COVID-19 ARDS [109]. Serum from patients with COVID-19 triggered NET release from control neutrocytes in vitro, and NETs released from SARS-CoV-2-activated neutrocytes promoted lung epithelial cell death in vitro, suggesting that NETs may mediate SARS-CoV-2-induced ARDS [99,112,113]. Angiotensin-converting enzyme 2, serine protease, viral replication, PAD4, and ROS were found to be associated with the activation of neutrophil-releasing NETs by SARS-CoV-2 [99,114]. Meanwhile, NETs can be used as a prognostic indicator for COVID-19. Multiple clinical studies have shown that circulating NET levels are associated with disease severity and affect clinical outcomes in patients with COVID-19, while NET levels are not associated with thrombosis development in patients, even though an association between NET formation and immunothrombosis is a well-established fact [113,115,116]. Similar to lung injury caused by other factors, NETs and their components contribute to the development of COVID-19 ARDS by activating downstream inflammatory pathways, damaging endothelial and pulmonary epithelial cells, and disrupting barrier function [77]. There is a higher rate of venous thromboembolism in severe COVID-19 ARDS than in other forms of ARDS [117]. NETs interact with platelets to cause a thrombose-inflammatory cascade, forming an immune thrombus that promotes lung damage in COVID-19, and NETs released from recruited neutrophils against SARS-CoV-2 do more harm than good in the lungs of patients with COVID-19 [118]. Neutrophils, which form NETs, also participate in the cytokine storm of COVID-19 [119,120]. The formation of NETs can lead to the production of excessive cytokines and chemokines, such as IL1β, IL6, IL8, IL10, TNF-α, and IFN-γ, which may trigger a cytokine storm leading to ALI, ARDS, and death [98,104,121]. At the same time, IL1β and IL8 also mediate the production of NETs, and thus may lead to uncontrolled progressive inflammation during cytokine storms [122,123,124]. The release of a large number of proinflammatory cytokines and chemokines can amplify the coagulation response and promote the formation of immune thrombosis [120]. In addition, Pei-Shan Sung et al. demonstrated that SARS-CoV-2-activated platelets produce COVID-19 extracellular vesicles through CLEC5A and TLR2 to induce NETosis and enhance thrombotic inflammation [125].

## 8. Clinical Research

At present, most clinical trials have been conducted on COVID-19 ARDS. Dornase alfa can act as a mucolytic agent and reduce NET levels in the lungs, thereby improving oxygenation and ventilation [126]. Andrew G. Weber et al. applied nebulized dornase alfa combined with salbutamol in five patients with ARDS who required mechanical ventilation due to SARS-CoV-2. After 7 days of treatment, FiO2 demand decreased, the condition improved in all five patients, and there were no deaths. The therapeutic effect of dornase alfa on inhibiting NETs and improving lung function was confirmed [126]. Similarly, the non-randomized trial of dornase alfa by Zachary M. Holliday’s group in the treatment of COVID-19 secondary ARDS also demonstrated the benefits of inhaled α-dornase, although the positive effect was limited to the time of administration [127]. In addition, the results of a clinical trial involving 20 patients with ARDS from COVID-19 treated with invasive mechanical ventilation suggest that metoprolol can also reduce neutrophil extracellular trap content and other markers of lung inflammation, reduce aggravated lung inflammation, and improve oxygenation without adverse effects, and is therefore a possible treatment strategy [128].

In addition, Xian Qiao et al. investigated the effects of intravenous vitamin C (HDIVC) injections on plasma-free DNA (cfDNA) and syndecan-1 in patients with sepsis-induced ARDS. The baseline levels of both biomarkers were reduced in the HDIVC group compared with the control group. The results showed that HDIVC could improve NETosis and endovascular calyx shedding [129]. In addition, a review of preclinical models and a meta-analysis of clinical studies on the efficacy of aspirin in ARDS suggested that acetylsalicylic acid (ASA) has a beneficial role in ARDS prevention and treatment, with the mechanisms of action including interference with neutrophil–platelet interactions, and the reduction of leukotrienes, neutrophil extracellular traps, and prostaglandins. One of these meta-analyses of three clinical studies also showed an association between ASA use and a reduced incidence of ARDS [130]. Currently, a clinical trial to verify that dornase alpha reduces the incidence of moderate-to-severe ARDS by reducing NETs in patients with severe trauma is also in phase III. However, the results of different clinical studies are contradictory, and the efficacy and application indications of ASA require further research [86]. The number of published clinical studies on the treatment of ARDS by targeting NETs is small, which means that treating ARDS by inhibiting or reducing NETs is still in an immature stage. The mechanism of action of NETs affecting ARDS and the treatment methods targeting NETs need to be further explored.

## 9. Treatment

Treatments targeting NETs have been validated by several studies and are considered to be a feasible treatment for ARDS, capable of alleviating pulmonary symptoms. Therapies targeting NETs include degrading already-formed NETs and inhibiting the formation of NETs (Table 2). The most common treatment for NET degradation is DNase Ⅰ, which degrades DNA components in NETs, thereby reducing the deposition of NETs, and is currently approved for clinical use without toxicity [131]. Moreover, multiple studies have demonstrated that DNase Ⅰ treatment in TRALI, VILI, and septic-induced ARDS mouse models can prevent NET accumulation, reduce lung injury, and improve survival rates [20,28,31]. After DNase I treatment, the levels of neutrophils and NETs were significantly reduced [95,132], the levels of inflammatory factors such as IL-1, IL-6, and TNF-α were decreased [26,132], and the levels of MPO activity, citrulline histone H3, and NE also dropped [106,132]. Paradoxically, however, studies have shown that DNase I treatment does not affect levels of inflammatory cytokines [95] and does not remove most of the NE and histone proteins, but only has a mild effect and partially reduces the damage [47]. Because the half-life of DNase Ⅰ limits its activity, nanomedics formed by fixing DNase Ⅰ on the surfaces of PDA nanoparticles can act more consistently and stably [106]. Another clinical trial of DNase I-coated melanin-like nanospheres also successfully reduced SARS-CoV-2-induced NETosis and plasma neutrophil counts, alleviated inflammation, and reduced mortality in ARDS models of sepsis [133].

Since PAD4 is the key to the formation of NETs, the inhibition of PAD4 can reduce histone citrullination and block NETosis [130,134]. PAD4 inhibitors Cl-amidine and GSK484, and selective inhibitors such as streptomycin, can reduce the levels of citrulline histone H3 (CitH3) and inflammatory factors, and reduce the formation of NETs, thereby reducing lung injury [9,135]. Treatment with the PAD2/PAD4 inhibitor YW356 has also been shown to reduce PAD activation and alleviate LPS-induced acute lung injury [79]. However, it is interesting to note that several studies suggest that targeting PAD4 has no benefit in improving sepsis survival, and that PAD2 inhibition leads to a decrease in NETosis and an overall improvement in prognosis [136,137].

As mentioned above, NE plays an important role in the formation of NETs, so inhibiting NE is another way to reduce the production of NETs. Using NE inhibitors can reduce the severity of lung injury [26,31]. The selective NE inhibitor GW311616A and the nanoparticle-mediated small molecule NE inhibitor Sivelestat have been shown to effectively inhibit the formation of NETs [9,62]. At the same time, since NE is also cytotoxic [62] and can mediate IL-1-induced excessive inflammation [126], treatment with irreversible elastase inhibitor alpha-1-antitrypsin (AAT) preconditioning improved prognosis in rat models with lung injury [138].

The inhibitors of some signaling pathways can also affect the formation of NETs; for example, inhibitors of p38 MAPK kinase can reduce NETs in ALI [95], and high-mobility histone B1 (HMGB1) can further drive NETs, so blocking it can prevent NET formation [36]. The hypoglycemic agent metformin can also reduce NETosis and lung inflammation by specifically inhibiting HMGB1, activating AMPK, and inhibiting the mTOR pathway [9,16]. Anti-PD-L1 antibodies affect the release levels of NETs by regulating neutrophil autophagy via inhibiting the PI3K/Akt/mTOR pathway [94].

Other therapies that target NETs, including the use of Protectin D1 (PD1) [139], iron-chelating agent Deferasirox [140], mesenchymal stem cells (MSCs) [141], osteopontin (OPN) [142], colchicine [143], disulfiram [144], Methoxyeugenol [145], anti-CLEC5A [146], Selinexor [147], etc., can inhibit or reduce the production of NETs. At the same time, the stimulation of the ear vagus nerve [148] and intravenous vitamin C [129] also block NETosis.

Overall, the treatment of targeted NETs is beneficial to reducing lung damage, reducing inflammation levels, improving prognosis, and improving survival rates in ARDS/ALI patients. However, more specific and effective treatments to reduce the production of NETs need to be further explored.

**Table 2 ijms-25-01464-t002:** Therapies targeting NETs.

Interventions		Effect	References
Clinical trial	Dnase Ⅰ	Removes DNA components from NETs, degrades NETs, and reduces NET deposition.	[106]
	Untravenous vitamin C	Inhibits NETosis, unspecified. ^1^	[129]
Observational Study	HMGB1 inhibitors, metformin	Improve efferocytosis and NET clearance.	[16]
Laboratory research	DNase I	Removes DNA components from NETs, degrades NETs, and reduces NET deposition.	[20,26,28,31,95,132,133]
	PAD4 inhibitors	Reduce histone citrullination and thus block NETosis.	[79,135]
	NE inhibitors	Inhibit the formation of NETs and reduce cytotoxicity.	[26,31,62]
	HMGB1 inhibitors	Block the signaling pathway and inhibit the formation and release of NETs.	[36]
	anti-PD-L1 antibodies		[94]
	inhibitors of p38 MAPK kinase		[95]
	Osteopontin (OPN)	Phosphorylation binds to histones with high affinity, thereby reducing cytotoxicity and reducing the formation of NETs.	[142]
	Protectin D1 (PD1)	Inhibit NETosis, unspecified.	[139]
	Deferasirox		[140]
	Mesenchymal stem cells (MSCs)		[141]
	Colchicine		[143]
	Disulfiram		[144]
	Methoxyeugenol		[145]
	Anti-CLEC5A		[146]
	Selinexor		[147]
	Stimulation of the ear vagus nerve		[148]

^1^ It is not clear in the literature in what form the interference reduces the contents of NETs.

## 10. Future Directions

At present, research on NETs is a hot topic, and the relationship between NETs and ARDS has been proven in recent years. However, there are still many problems regarding NETs that need to be further explored and solved.

For example, for the formation of NETs, the classification of NETosis includes not only the recognized NOX-dependent and independent NETosis, but also the newly proposed ROS-dependent NETosis with the rapid release of mitochondrial DNA and granular protein [15]. This suggests that there are many details about the mechanism of NETosis that have yet to be discovered and elucidated. PAD4 plays a key role in the formation of NETs, but it is not indispensable, and its role is not limited to citrullinated histones, an area which requires more in-depth exploration and interpretation. The relationship between platelets and NETs also needs to be further researched. Activated platelets can induce the formation of NETs through TLR4 and its downstream signaling pathway, and NETs act on lung tissue and cause coagulation and thrombosis, which seems to become a vicious cycle.

In terms of the impact of NETs on the lungs, when NETs and their components cause lung tissue damage in various forms, does anyone play a leading role in promoting the progression of ARDS? Or are there other, more critical, NET pathways or nodes that need to be further explored? At present, it has been confirmed that there is a correlation between NETs and the severity of ARDS, that NETs can reflect the prognosis of ARDS, and that the number of NETs will change dynamically during the clinical course of ARDS, but we still do not know whether there is a causal relationship or a feedback effect between the two.

Finally, on the therapeutic side, targeting NETs to alleviate the development of ARDS has been shown to be a viable measure, and DNase I, which has the effect of clearing DNA, a key component in NETs, has been approved for clinical treatment. However, according to the results of multiple DNase I related studies and the mechanism of action of NETs, more comprehensive and effective drugs still need to be developed. Preventing the formation and release of NETs may be a better choice, but it is also important to take into account the role of NETs in removing pathogens, and to prevent NETs, the inhibition of which can cause ARDS infections, from worsening the disease due to uncontrolled infection.

## 11. Conclusions

This review highlights the strong link between NETs and ARDS. In the process of ARDS, neutrophils accumulate in and infiltrate the lungs to generate NETosis, and the NETs produced destroy the lung endothelial cell barrier, causing tissue damage, forming immune thrombosis, and aggravating the condition of ARDS. The production of NETs and the release of contents that affect the surrounding tissues constantly stimulate each other, forming a vicious circle, leading to the continuous formation of NETs. At the same time, NETs can also cause cytokine storms and affect the body’s immune balance, further promoting the development of the disease. However, although there has been much research on the formation of NETs in ARDS, the mechanism affecting disease progression, and the corresponding treatment methods, specifically exploration and interpretation, still need further study. Therefore, we need further research on the interaction of NETs with ARDS in order to obtain new treatment strategies for ARDS.

## Figures and Tables

**Figure 1 ijms-25-01464-f001:**
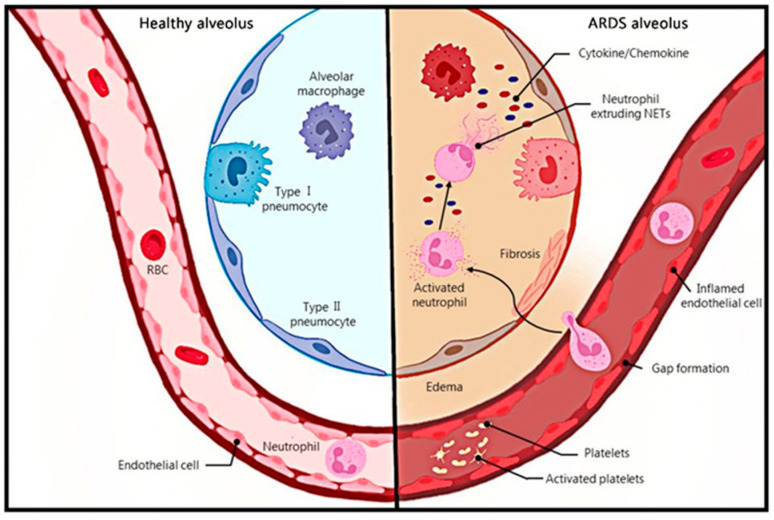
Contrast between healthy alveolus and ARDS alveolus. Neutrophils infiltrate in ARDS lung tissue, and activated neutrophils release NETs.

**Figure 2 ijms-25-01464-f002:**
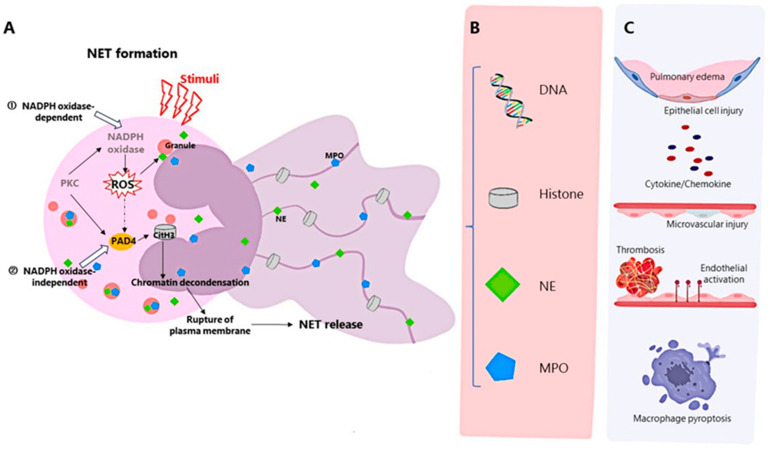
Formation of NETs, components of NETs and mechanisms affecting ARDS. (**A**) The formation of NETs can be divided into NADPH oxidase-dependent and independent formations. ① Activation of PKC leads to activation of NADPH oxidase and the production of ROS. ROS stimulates MPO and NE to migrate to the nucleus, NE triggers histone degradation, and MPO collaborates with NE to promote chromatin decondensation. ② Activated PAD4-mediated histone citrullination leads to chromatin decondensation [9]. (**B**) The major components of NETs include DNA, histones, NE and MPO. (**C**) NETs and their components can cause pulmonary tissue edema, damage alveolar epithelial cells and microvessels, cause the formation of immune thrombosis and activation of endothelial cells, and mediate the pyroptosis of macrophages and the release of downstream cytokines.

## Data Availability

Not applicable.

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
