# Peer review of "A Narrative Review: The Role of NETs in Acute Respiratory Distress Syndrome/Acute Lung Injury"

_ijms, 2024, doi:10.3390/ijms25031464_

Round 1

Reviewer 1 Report

Comments and Suggestions for Authors
  • The main question addressed is the role of neutrophil extracellular traps (NETs) in the pathogenesis, progression, and potential treatment of Acute Respiratory Distress Syndrome (ARDS). The topic is highly relevant and original in the field of respiratory medicine. It addresses a specific gap by consolidating current understanding of NETs in ARDS, a crucial aspect of the disease that has significant implications for treatment and prognosis. This review adds to the subject area by providing a comprehensive and detailed examination of NETs in ARDS. It stands out by discussing the molecular mechanisms in depth and exploring potential therapeutic approaches, which have not been as thoroughly covered in previous publications. It is recommended that the authors clearly declare the manuscript as a narrative review in the title for immediate clarity on the scope and methodology. As this is a narrative review, the methodology primarily involves the synthesis of existing literature. To enhance the review, the authors could consider including a more systematic approach to literature selection and analysis. Expanding on specific signaling pathways and causality in the formation of NETs in ARDS would enhance the depth of the review. This would involve clearly stating the inclusion and exclusion criteria for studies and perhaps providing a flowchart of the literature review process. Emphasizing the clinical relevance of these findings could aid in translating this knowledge into clinical practice or influencing future research directions. The conclusions are consistent with the evidence and arguments presented and successfully address the main question posed. They provide a coherent understanding of the role of NETs in ARDS, supported by a wide range of studies. The references are appropriate, current, and relevant. They span a range of studies that provide a robust foundation for the review's claims.
  • The figures, particularly the contrast between healthy and ARDS alveoli, effectively illustrate key concepts. However, additional diagrams depicting the molecular pathways involved in NET formation and their role in ARDS could further enhance understanding. Tables summarizing key studies and their findings related to NETs in ARDS could be beneficial for readers, providing a quick reference to crucial data.

    General Suggestions:

  • Clarify in the title that this is a narrative review to set appropriate expectations for the readers.

  • Include a section summarizing potential future research directions or unanswered questions in the field to guide subsequent studies.
  •  
  • These suggestions and comments are intended to further strengthen the manuscript and aid in conveying its significant contributions to the understanding of NETs in ARDS.

Reviewer 2 Report

Comments and Suggestions for Authors

The authors review data regarding the impact of NETosis in the development of ARDS from various etiologies, mechanisms of effect, and possible therapies. The review is thorough and generally in-place. The NET concept is about 20-years old, and continues to evolve, with exciting research and possible clinical implications.  

However, I have some serious concerns regarding the manuscript:

1. I believe that there should be a better design to the manuscript, as there is a lot of confusion between preclinical and laboratory data, clinical data, and therapeutic implications, as well as between various mechanisms and models of disease. This leaves the reader quite confused, and the manuscript is hard to follow.

2. I believe that there should be better references to the clinical implications of the scientific data. It is very not organized currently.

3. There are some novel studies that correlates NET with thromboinflamation, which are not covered in the manuscript.

4. The tables and figures does not add meaningful insights or data, and are only confusing (such as the Table 2 of possible management - why are some interventions considered inhibitory and others not?)

Comments on the Quality of English Language

some minor typos should be corrected (TRAIL VS. TRALI)
